# Ubiquitin Proteasome System and Microtubules Are Master Regulators of Central and Peripheral Nervous System Axon Degeneration

**DOI:** 10.3390/cells11081358

**Published:** 2022-04-15

**Authors:** Daniele Cartelli, Guido Cavaletti, Giuseppe Lauria, Cristina Meregalli

**Affiliations:** 1Neuroalgology Unit, Department of Clinical Neurosciences, Fondazione IRCCS Istituto Neurologico Carlo Besta, Via Celoria 11, 20133 Milan, Italy; giuseppe.lauriapinter@istituto-besta.it; 2Experimental Neurology Unit and Milan Center for Neuroscience, School of Medicine and Surgery, University of Milano-Bicocca, Via Cadore 48, 20900 Monza, Italy; guido.cavaletti@unimib.it; 3Department of Medical Biotechnology and Translational Medicine, University of Milan, Via Vanvitelli 32, 20129 Milan, Italy

**Keywords:** axon regeneration, chemotherapy-induced peripheral neuropathies, dying back, microtubules, Parkinson’s disease, ubiquitin proteasome system

## Abstract

Axonal degeneration is an active process that differs from neuronal death, and it is the hallmark of many disorders affecting the central and peripheral nervous system. Starting from the analyses of Wallerian degeneration, the simplest experimental model, here we describe how the long projecting neuronal populations affected in Parkinson’s disease and chemotherapy-induced peripheral neuropathies share commonalities in the mechanisms and molecular players driving the earliest phase of axon degeneration. Indeed, both dopaminergic and sensory neurons are particularly susceptible to alterations of microtubules and axonal transport as well as to dysfunctions of the ubiquitin proteasome system and protein quality control. Finally, we report an updated review on current knowledge of key molecules able to modulate these targets, blocking the on-going axonal degeneration and inducing neuronal regeneration. These molecules might represent good candidates for disease-modifying treatment, which might expand the window of intervention improving patients’ quality of life.

## 1. Introduction

Nervous system functioning depends on the intricate arborisation of neurons and the interconnection they establish with each other and with glial cells during development, some of which are continuously re-modelled during life. Axon degeneration causes the detachment of neurons from the targets, with the consequent loss of the regulation of physiological processes, and it underlies many neurological disorders. Axon degeneration is not blocked by genetic or chemical inhibitors, indicating that it is regulated differently from programmed neuronal death [1]. The analyses of programmed cell death identified different intracellular pathways that can lead to neuronal loss, as the increase of oxidative stress or re-activation of factors regulating the cell cycle [2]; nevertheless, a causal relation between these events and the proper neurodegenerative state is still under debate.

Various pathways mediate the removal of axons. One of the simplest to explore experimentally is Wallerian degeneration (WD) which derives from the transection of axons [3], and although it was first described almost two centuries ago, its molecular mechanisms have been the object of clarification during the last three decades [4,5]. Some of the molecular pathways are shared with the Wallerian-like degeneration termed “dying back”, which characterizes the loss of neuronal populations in different neurodegenerative disorders affecting central and peripheral nervous systems like, for example, Parkinson’s disease (PD) and peripheral neuropathies. Typically, dying back degeneration starts distally and progresses toward the soma [6]. Both these conditions share several reasons for axon degeneration, including that axon pathology is length dependent, the extreme vulnerability to metabolic demand for long distance transport, and the fact that they may derive from exposure to environmental neurotoxins and drugs. A further commonality between PD and peripheral neuropathies is the fact that in the central nervous system there is no evidence of long-range axon regeneration and, sometimes, peripheral nerves even fail to regrow [7]. Last but not least, there are hints about the presence of peripheral fiber pathology in PD patients [8], sustaining the hypothesis of common pathological mechanisms, such as the activation of similar intracellular players, among which two deserve particular attention: microtubules (MTs) and the ubiquitin-proteasome system (UPS).

### 1.1. Microtubules and Axonal Transport

MTs are polar structures constituted of α/β tubulin heterodimers that alternate slow polymerizing and rapid shrinking phases, a behavior called “dynamic instability” which is more pronounced at the so-called plus-end. In axons, MTs seem to be uniformly plus-end out-oriented, meaning the cellular cargo is transported from the cell to the periphery with more ATP-powered growth, an organization that promotes neuronal polarization [9]. Conversely, in dendrites, MTs are non-uniformly aligned and this mixed orientation drives the minus-end directed dyneins to deliver dendrite-specific cargo [10].

In neurons, MTs heterogeneity is further increased by the combination of different tubulin isoforms, post-translational modifications and MT-associated proteins, which have the potential to generate surface patterns associated with diverse functions, correlating with MT stability (i.e., more stable MTs survive for a longer time) and being associated to different neuronal subregions (Figure 1) [11].

For example, the dynamic tyrosinated MTs are enriched in highly mobile neuronal regions, as presynaptic terminals and dendritic spines [12], whereas the axonal shaft is enriched in detyrosinated/Δ2 tubulin and acetylated MTs, modifications that are variously related to the recruitment of interacting proteins and change in MT stability [13], to protection from severing enzymes [14] and axonal regeneration. Furthermore, they regulate both kinesins, plus-end oriented molecular motors and dyneins velocities and processivity [15]. Acetylation of lysine 40 on α tubulin deserves particular attention, since it is the only known luminal MT modification, and it acts as a molecular clock regulating the aging of MTs and their self-repairing capability [16], two important features for structures lasting for the entire life of a neuron.

Axonal transport is the bidirectional trafficking of diverse organelles and cargoes. Due to their preferential direction of movement and the uniform orientation of MTs, in axons, kinesins and dyneins are respectively in charge of the anterograde (from soma to tips) or retrograde transport of proteins and organelles [17]. Whether the axonal transport block is the direct consequence of MT defects or other cellular damaging events remains elusive, but it is a crucial event in several neurodegenerative disorders [18].

Therefore, targeting specific pathways which regulate axonal transport could become a potential therapeutic strategy. One of the most promising targets is histone deacetylase 6 (HDAC6), which removes the acetyl groups from the polymerized MTs [19,20], mediating their activity and the regulation of molecular motors and axonal transport. Its genetic or pharmacological modulation shows protective actions in both animal and human experimental models of different neurodegenerative disorders [21].

### 1.2. Ubiquitin-Proteasome System and Protein Quality Control

The UPS and lysosome-mediated protein degradation via aggresome formation/autophagy are two major degradative pathways of proteins and, since about 30% of newly synthesized proteins are misfolded under physiological conditions, the precise co-operation of these systems is fundamental for cell functioning. In the UPS, proteins to be degraded are ubiquitinated by specific enzymes, and the substrate specificity is determined by a family of more than 600 proteins, the E3 ubiquitin ligases. The other two components of the UPS are the deubiquitinating enzymes and the 26S proteasome, a multi-subunit proteinase complex, which ultimately degrades the ubiquitinated substrates (Figure 2, upper panel) [22]. The structural–functional relationships of the UPS components have not been fully identified yet, albeit recent works have shed light on how the impairment of ubiquitin-mediated degradation can lead to toxic protein aggregation in age-related degenerative diseases [23,24,25].

Among the three functional components of UPS, the E3 ligase proteins represent the link with the other cell structures and pathways, and gain- or loss-of-function mutations of a number of E3 genes affect normal neuronal functioning and lead to axon degeneration and nervous system disorders. For example, the PAM-Highwire-Rpm-1 (Phr1) E3 ligase is an effector of the axonal destruction pathway and its loss favors the survival of damaged axons in both the central and peripheral nervous systems [26]. In agreement with this observation, increased activity of an E3 ligase limits the availability of the axonal maintenance factor nicotinamide mononucleotide adenylyltrasferase 2 (NMNAT2), thereby initiating degeneration [27].

### 1.3. Axon Degeneration

WD and the dying back process converge on a programmed pathway of injury signaling and cytoskeleton degradation which results in axonal degeneration [28]. Studies have clarified that the loss of one peculiar axonal protective molecular player, the enzyme NMNAT involved in the biosynthesis of NAD+ [4], is crucial to axonal degeneration. Since the axonal isoform of NMNAT, NMNAT2, is constantly replenished by anterograde axonal transport, its deficiency due to axonal transport block or other conditions such as increased proteasome activity can induce axonal degeneration.

Conversely, the loss of function of sterile alpha and TIR motif containing 1 (SARM1), an injury-activated NAD+ cleavage enzyme, delays axon degeneration and has neuroprotective capabilities through various mechanisms in some [29,30,31] but not all neurodegenerative disorders [32]. SARM1 function or loss of NMNAT2 enzymatic activity can trigger the activation of dual leucine kinase (DLK)/MAPK signaling that eventually induces calpain-mediated axonal degeneration [33]. It has been reported that the inhibition of DLK activation could delay axonal degeneration after both axotomy and chemotherapeutic treatment [33]. In agreement with this findings, a recent study showed that the in vivo overexpression of the DLK-interacting protein FK506-binding protein 8 (FKBP8) delayed the progression of axon degeneration and suppressed neuronal death through the regulation of DLK stability via the UPS and lysosomal protein degradation pathways [34]. Once the axon is damaged, NMNAT2 reduction or SARM1 activation induces calcium influx that starts the calcium-dependent calpain degradation of the axon, in which the active role of SARM1 is confirmed in human derived sensory neurons models of traumatic and neurotoxic degeneration [35]. Other studies revealed that the cytoskeleton, and especially the MTs, are involved in the active phases of axonal destruction (Figure 2, lower panel). For example, one study demonstrated that MT fragmentation is the earliest detectable event during WD, and that both genetic and pharmacological inhibition of UPS delayed degenerative events both in vitro and in vivo [36]. In agreement with the crucial role of UPS, the E3 ligase Phr1 tunes the levels of NMNAT2 promoting axonal self-destruction [26,27]. Other studies showed the importance of MTs, as disorganized MTs cause the formation of retraction bulbs and the pharmacological inhibition of MTs leads to regeneration failure [37], whereas direct MT stabilization promotes axon regeneration in vivo [38].

As an example of how diseases with different etiologies and different types of stress that initiate axon loss can share the same mechanisms of axon degeneration, we will focus on PD and peripheral neuropathies to show some convergences onto common pathways, similar length-dependent axonal pathology and susceptibility to drugs.

## 2. Parkinson’s Disease

Disorders affecting the central nervous system require the activation of widespread processes whose molecular mechanisms are still elusive, and neurodegenerative diseases can be roughly divided into dementias, including Alzheimer’s disease, or motor disorders, such as PD, Huntington’s disease or Amyotrophic Lateral Sclerosis. This last class usually derive from the degeneration of long-projecting neuronal populations, such as dopaminergic neurons in PD, which is the second most common neurodegenerative disorder, affecting 2–5% of people over 60s worldwide; it is characterized by bradykinesia, resting tremors and rigidity, symptoms that arise from the lack of dopamine in the striatum due to the death of the dopaminergic neurons in the substantia nigra [39]. The identification of druggable pathways leading to disease-modifying treatments remains an unmet clinical need. The inhibition of the mitochondrial respiratory chain and the UPS failure are amongst those acknowledged as playing key roles [40]. Indeed, ubiquitin-immunopositive inclusions in the Lewy bodies, which are a neuropathological hallmark, can be found in the brain of PD patients [41] and impaired proteasomal activity was reported in the substantia nigra [42]. On the other hand, dopaminergic neurons have very long projections whose average arborisation reaches 4.6 m in humans [43], making them particularly dependent on proteins and organelles supply and, thus, on cytoskeletal integrity and axonal transport, which has been clearly associated to the early phase of PD [44]. The nigrostriatal system displays dying-back degeneration that resembles WD, with MT fragmentation as the first detectable damage [36]. Similarly, MT disorganization has been extensively associated to the driving phase of axon degeneration in PD (for a detailed review see [45]). However, not all experimental models of PD require the participation of actors playing a pivotal role in WD, such as SARM1 or NMNAT2; indeed SARM1-driven axonal degeneration has been reported only in a single neurotoxic model of PD [46].

Studies based on PD-inducing neurotoxins (as rotenone, 1-methyl-4-phenyl-1,2,3,6-tetrahydropyridine or MPTP, 6-hydroxydopamine and 2,5 hexanedione) demonstrate that dopaminergic neurons are susceptible to MT-targeted drugs more than other neuronal populations [47] and that derangements of MTs are precocious events that precede and may induce trafficking alterations and mitochondrial failure [48,49]. The common defects resulting from these toxins are the early loss of dynamic MTs, the accumulation of disorganized and fragmented acetylated MTs and the block of axonal transport. The early MT destabilization is not only a by-product of the on-going degeneration since the MT-stabilizer epothilone D is neuroprotective in MPTP -treated mice [50]. This neurotoxin likely activates the same pathway involved in WD, as its administration in Wld(S) mice, a strain resistant to Wallerian-like axonal degeneration, did not induce an early axonal loss [51]; similarly, Wld(S) mice treated with 6-hydroxydopamine showed a delayed axonal degeneration [52].

Many pathological mutations have been reported in different genes that are causally linked to PD, but only two of them account for the vast majority: PRKN, encoding for parkin and representing the major cause of juvenile onset PD, and LRRK2 that encodes for leucine-rich repeat kinase 2 (LRRK2). Both proteins are enzymes involved in many intracellular processes and displayed the ability to interact with actin and tubulin, mediating MT acetylation and having an impact on the transport of mitochondria [53,54,55,56]. Interestingly, pathological mutants or protein absence corrupts these processes but these could be rescued by the modulation of MT acetylation [53] or by the use of MT-targeted drugs [57].

Genetic evidence suggests the involvement of UPS dysfunction in PD. Pathological mutations have been reported both in the E3 ligase parkin [58], and in the deubiquitinating enzyme ubiquitin C-terminal hydrolase L1 [59], encoded by UCHL1. Loss-of-function parkin mutants fail to interact with α-synuclein, leading to its aggregation, ubiquitination and accumulation into Lewy bodies in humans [60], and to an age-dependent degeneration of the nigrostriatal pathway accompanied by the appearance of motor deficits in mice [61]. Parkin is also involved in the clearance of mitochondria and MTs. Studies demonstrate that the Parkin/phosphatase and tensin homolog (PTEN)-induced kinase 1 (PINK1) pathway regulates both the transport of functional and damaged mitochondria [54] and mitophagy [62], the process through which the dysfunctional mitochondria are selectively removed. Furthermore, parkin interacts with and promotes the degradation of tubulin by UPS [63], highlighting the extent to which MTs and UPS are intertwined and how E3 ligase can represent a point of convergence between these two systems. Thus, the evidence here reported shows how the alteration of similar pathways can lead to axonal degeneration in PD as well as in WD, but that protective mechanisms may be different.

## 3. Peripheral Neuropathies and Chemotherapy-Induced Peripheral Neuropathy

Neuropathies represent a heterogeneous entity, as they can be classified as motor, sensory or mixed, such as Charcot-Marie-Tooth disease, and are characterized by loss of muscles and weakness, altered touch sensation and sensitivity to warm and cold; symptoms may also include nociceptive and neuropathic pain. Peripheral neuropathies can be either acquired or inherited, whose central abnormality is axon degeneration. Acquired peripheral neuropathies are amongst the most common neurological disorders worldwide [64] and are a major complication of antineoplastic therapies, affecting at least 50% of patients [65]. They experience a number of symptoms including numbness, paresthesia, neuropathic pain and weakness [66]. The most common cause of chemotherapy-induced peripheral neuropathy (CIPN) are platinum-based compounds, anti-tubulin drugs and proteasome inhibitors [67]. These compounds share a common pathway triggering an early axonal degeneration. Similar to dopaminergic neurons, long sensory nerves are highly susceptible to axonal transport impairment as a causative role contributing to sensory axon loss, which is correlated with the typical dying back and length-dependent presentation of CIPN [68]. Paclitaxel (PTX) is among the most used antineoplastic drugs causing CIPN. The axonal degeneration induced by PTX might be due to interruption of the axonal transport or to the block of regeneration due to the over-stabilization of re-growing neurites [69]. In addition, other studies have demonstrated that PTX-induced post-translational modifications of MTs can interfere with motor proteins binding and motility [70,71]. PTX is a paradigmatic molecule to show how interconnected are the pathways we are discussing. Indeed, its action on the MT system is well described, while the resistance of the mouse mutant Wld(S), characterized by axonal degeneration delay in a variety of disorders, to PTX-induced neuropathy [72] suggests possible common mechanisms between WD and PTX-induced CIPN.

In line with these observations, the involvement of NMNAT2 and SARM1, the two WD actors required for axon survival and degeneration, respectively, has been reported in several models of peripheral neuropathies. For example, reduced transport of NMNAT2 may increase distal axon sensitivity to degeneration due to decreased NMNAT2 levels and is considered common to many peripheral neuropathies [69]. Other studies support the hypothesis of a shared mechanism between WD and CIPN, since both vincristine (VCR) and bortezomib (BTZ) activate a common SARM1–dependent axon degeneration pathway [73]. Interestingly, a recent paper proposed a novel function for SARM1 in sensory neurons, as a negative regulator of axonal cytoskeleton dynamics and collateral branching; indeed, SARM1 depletion leads to increase of MT polymerization dynamics in cultured neurons and to an increased branching of cutaneous sensory innervation in skin of SARM1 ko mice [74]. Modifications of cytoskeletal proteins are described in several peripheral neuropathies and they might interfere with cytoskeleton assembly [75] and thus axonal transport [76]. Disruption of MT dynamics can impair the axonal transport of several elements, which is the case in taxanes and vinca alkaloids compounds, resulting in axonal damage [77,78] and CIPN.

The pivotal role of MT system dysfunctions in the development of CIPN is highlighted by the tubulin-directed agents and strengthened by evidence arising from drugs that target other systems, such as BTZ and cisplatin (CDDP). BTZ is the first proteasome inhibitor introduced for the treatment of hematological malignancies and solid tumors, which acts by inhibiting the chymotrypsin-like enzymatic activity of the 26S proteasome (Figure 2, lower panel), inducing an accumulation of the polyubiquitinated proteins [79].

BTZ causes a sensory-predominant peripheral neuropathy and, although it is structurally unrelated to tubulin drugs, the axonopathy is due to MT over-stabilization in tissue culture cells [80] and accumulation of tubulin in DRG neurons with a time-dependent reduction of mitochondria transport [76]. BTZ can also induce alterations in MT dynamics and molecular transport in human induced pluripotent stem cell-derived sensory neurons [81]. Recent studies demonstrated, both in vitro and in vivo, that BTZ increased tubulin polymerization, suggesting that an alteration of tubulin dynamics and chronic proteasome inhibition contribute to BTZ-induced peripheral neurotoxicity [82]. Furthermore, BTZ was shown to cause in vitro and in vivo pathological accumulation of Δ2 tubulin in sensory neurons, while a reduction of Δ2 tubulin rescued both mitochondria trafficking defects and axon degeneration [83].

As reported for parkin, BTZ affects mitochondria quality inducing mitotoxicity and proteotoxicity [84], and it also promotes the mitochondrial accumulation of the parkin-interactor PINK1 leading to mitophagy [85]. Accordingly, PINK1 expression ameliorates thermal hypersensitivity in PTX-treated Drosophila larvae through the suppression of PTX-induced mitophagy [86]. This evidence suggests a close interrelationship between all these systems.

In addition, some in vitro studies on CDDP neurotoxicity, a chemotherapeutic drug that interferes with DNA repair inducing apoptosis of cancer cells, showed impaired fast axonal transport associated with light morphologic alterations [87]. Interestingly, CDDP treatment in Drosophila resulted in a large number of stationary mitochondria caused by extended pausing during axonal motility [88]. In a mouse model, the HDAC6 inhibitor ACY-1215 (ricolinostat), which acts on acetylated tubulin, was able to prevent mechanical allodynia, rescue mitochondrial bioenergetic contents and restore intraepidermal nerve fiber density in CDDP-treated mice [89]. Therefore, as we already described in PD contexts, the modulation of MT acetylation might be a potential candidate for reverting the clinical and neuropathological defects associated to some CIPN. All this data highlights that axon destruction programs converge on common pathways although the eliciting events might differ, and explain how the study of particular pathologies can be relevant to our understanding of the general process. 

## 4. Regenerative Potentialities

Strategies to manipulate NMNAT2 or SARM1 are particularly attractive for counteracting on-going degeneration. NMNAT2 is an endogenous survival factor able to sustain labile axons. Its expression level needs to be maintained above a certain threshold to protect neuronal processes from degeneration [90], which can be done by its continuous replenishment through active anterograde transport or by limiting its degradation by UPS. In a Drosophila injury model, it was demonstrated that the highly conserved E3 ubiquitin ligase, Highwire, strongly inhibits WD through the modulation of its downstream target, NMNAT2 [27]. In agreement with the role of E3 ligase in maintaining and promoting axonal survival, the UPS related neural precursor cell expressed developmentally down-regulated protein 4 (Nedd4) induces the degradation of PTEN and promotes axon branching and regeneration [91]. The crucial role of PTEN in blocking axonal regeneration has been confirmed in a very recent paper showing that PTEN deletion induces neuron hypertrophy and regeneration [92]. Nedd4 is also a critical regulator of excitability of nociceptive neurons [93]; thus, it has the potentiality to induce both axon regeneration and pain relief.

In recent years, SARM1-depletion has gained attention for its ability to protect from axonal degeneration, and the possibility of pre-treating patients with SARM1 inhibitors before they receive chemotherapy might offer a potentially successful strategy to prevent CIPN. Albeit starting from different engaging mechanisms, VCR and BTZ converge toward a core axonal derangement program that can be inhibited by the expression of candidate genes which deplete SARM1 [73]. In keeping with this observation, SARM1-deficient mice are resistant to Wallerian-like neuropathy induced by PTX or a high fat diet [94]. In addition, pharmacological SARM1 inhibition prevented axon structure alterations and provided partial protection of the axonal function in PTX-treated animal models [95].

The modulation of MT functioning for blocking axonal degeneration or inducing their regeneration made MTs a potential druggable target for disease-modifying treatments [96]. The first hypothesis was to use MT-stabilizers like epothilone D in murine models of PD [50] and Alzheimer’s disease [97], but their side-effects, i.e., the induction of peripheral neuropathy, limited their use in patients. Recently, new possibilities are under investigation, such as the use of peptides specific for MT lattice and regulation of post-translational MT modifications. Peptides mimicking tubulin sequence at dimer–dimer interface can induce changes in the longitudinal or lateral contacts, thus having the potentiality to regulate MT stability without causing cell resistance or pathological changes. Davunetide (NAPVSIPQ) is one of the first experimentally used peptides and, due to its very high efficiency in inducing recovery from transport defects and axon damage in different neurodegenerative models, it underwent a clinical trial for the treatment of patients affected by Progressive Supranuclear Palsy, which however did not prove to be efficacious [98]. The regulation of MT post-translational modifications has different possible readouts such as the impact on the MT lattice itself or on MT-interacting protein and molecular motors, thus modulating MT stability and axonal transport. For example, the tubulin-tyrosine ligase increases the level of tyrosinated MTs at axonal injury sites, mediating retrograde organelle transport and starting the signaling essential to axon regeneration [99]. Similarly, the increase of tyrosinated MTs induces axon regeneration in DRG neurons after injury in vivo [100] and the inhibition of MT detyrosination by the administration of cnicin or parthenolide accelerates regeneration and functional recovery of axons in vivo [101]. Moreover, the accumulation of Δ2 tubulin has been associated to BTZ-induced neuropathy and its modulation by multiple enzymes reduces axonal degeneration [83]. As already remarked on, a special mention goes to tubulin acetylation and the enzymes that mediate it, because it may represent an accessible target for interventions aimed at interfering with MT functional organization, axonal transport and axon regrowth. For example, sirtuin-2 deacetylates tubulin increasing the resistance to axonal degeneration in Wld(S) mice [102] and its active role in the progression of PD and Alzheimer’s disease has been suggested [103]. On the other hand, HDAC5 is an injury-regulated enzyme which modulates tubulin acetylation at growth cone promoting axon advancement and regeneration [104].

The most promising target is HDAC6, which proved its ability in rescuing axon degeneration in many experimental models, acting on three interconnected pathways such as tubulin acetylation, axonal transport and autophagy of misfolded proteins [21]. It has been demonstrated that HDAC6 is up-regulated in a spinal cord injury model, and that its inhibition is able to increase tubulin acetylation and organelle transport, thus restoring the autophagic flux and inducing axonal elongation [105]. Through the same mechanisms, its inhibition is neuroprotective in a drug-induced PD animal model [106]. Moreover, both genetic and pharmacological regulation of MT acetylation could rescue axonal transport defects and motor deficits in the LRKK2 mutants-induced drosophila model of PD [50] and in the in vitro model of Huntington disease [107]. As previously reported, the administration of ricolinostat in CDDP-treated mice could restore axonal defects and induce the regeneration of sensory terminal nerve endings in the skin [89], ameliorate the behavioral effects and improve the cognitive decline associated to Alzheimer’s disease transgenic mice [108]. Furthermore, HDAC6 inhibitors demonstrated the beneficial effects in preclinical models of the axonal form of Charcot-Marie-Tooth [109], removing aggregated protein, rescuing organelles transport and promoting axon regeneration. This evidence provides a rationale for the use of HDAC6 inhibitors as a therapeutic strategy for different, although not all, disorders here described.

## 5. Closing Remarks

We have presented updated results showing how different pathways co-participate in the axon destruction and how their modulation has the potentiality to induce neuronal regeneration. A strong candidate is the pre-treatment of patients that have to undergo chemotherapy with inhibitors of SARM1, or with multiple antineoplastic agents and/or with genetic regulators of convergent pathways (i.e., miRNA modulators), solutions already used in clinical oncology that might widen the window of intervention [110]. Unfortunately, no one can predict if an individual will develop PD or other neurodegenerative disorders; thus, no preventive treatments are allowed. Among all the possible targets that can prompt axon regeneration, HDAC6 is a promising candidate as it is easily accessible to both pharmacological and genetic points of views, and its action impacts multiple systems. Over the next years, researchers and clinicians have to identify which is the best target in the different pathological or therapeutic contexts, as well as how and when to modulate it.

## Figures and Tables

**Figure 1 cells-11-01358-f001:**
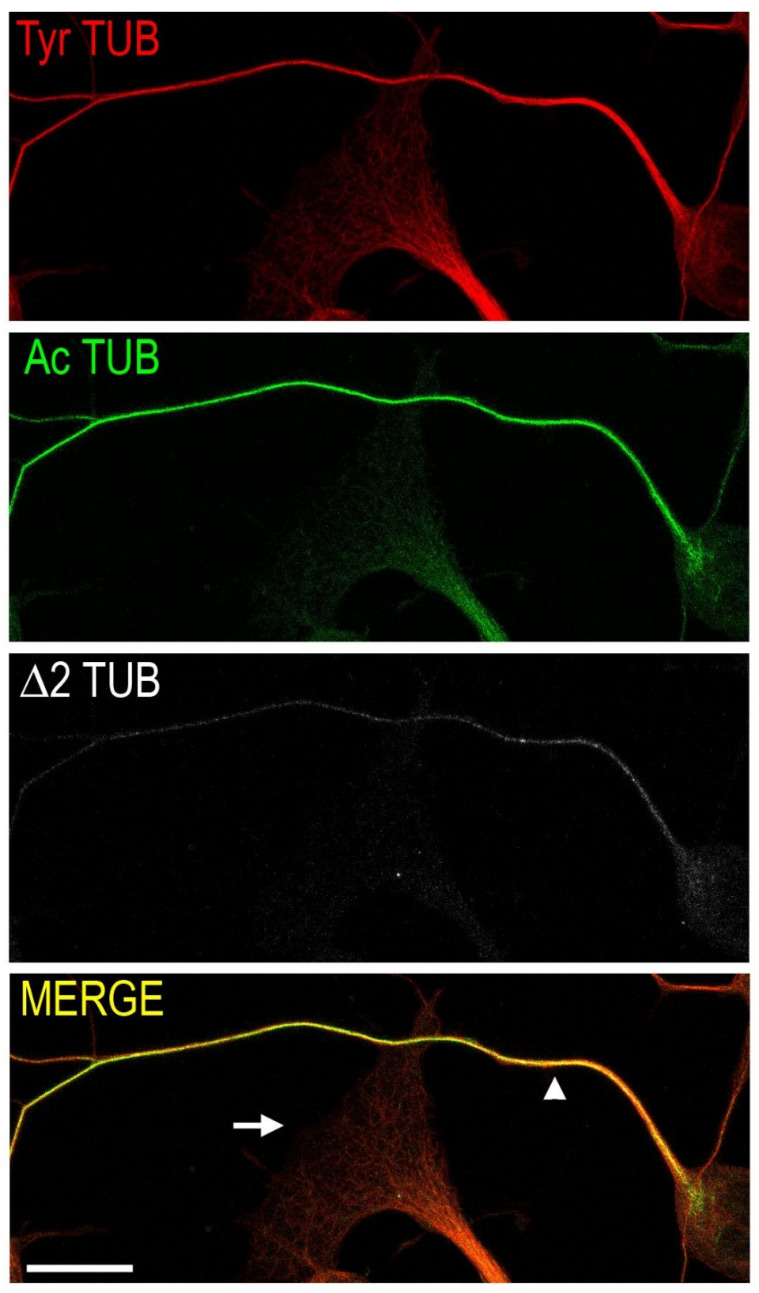
Confocal micrograph of cultured neurons, showing a conventional distribution of different tubulin post-translational modifications along the axon (arrowhead) and inside an enlarged terminal (arrow). Tyrosinated tubulin (Tyr TUB, red) is present in the axon and constitutes the microtubule network at the terminal; acetylated tubulin (Ac TUB, green) is enriched along the axon and barely present at the tip; Δ2 tubulin (Δ2 TUB, white) is detectable only in the axon shaft showing a proximal-to-distal distribution. Scale bar = 15 µm.

**Figure 2 cells-11-01358-f002:**
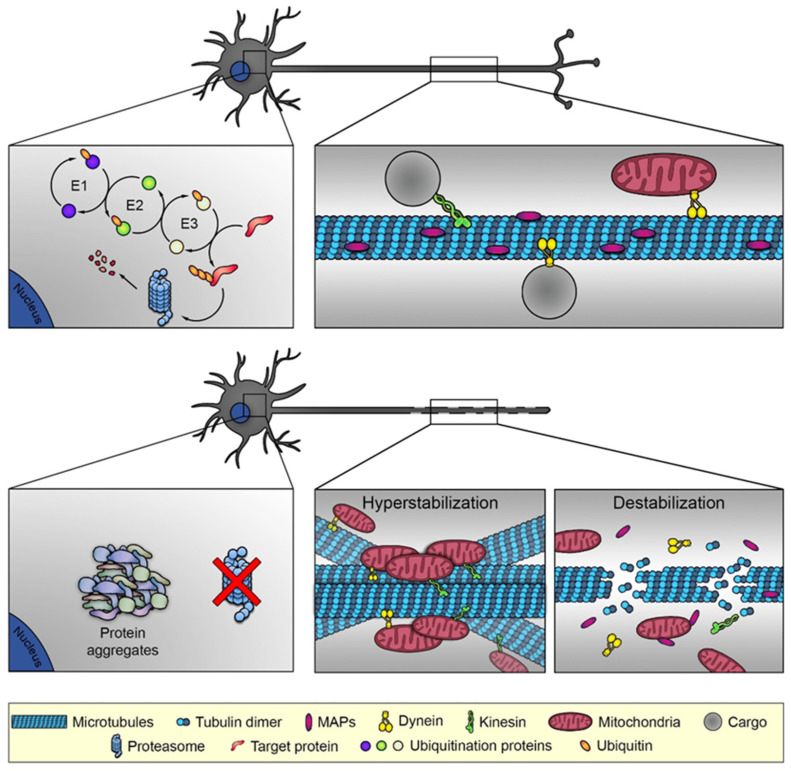
In healthy neurons (**upper panel**), ubiquitin-proteasome system is fully active and degrades the unfolded proteins. Not only, the microtubule organization is conventional, as depicted in Figure 1, and sustains the microtubule-dependent functions as the axonal transport of mitochondria or other cargos. In damaged neurons (**lower panel**), the ubiquitin-proteasome system is blocked, promoting the accumulation of misfolded proteins that induce aggregation, toxicity and axonal degeneration. Microtubule-dependent axonal destruction may be derived from two different conditions: microtubule hyperstabilization, which induces microtubule bundling and block of axonal transport, and microtubule destabilization, with consequent release of microtubule interacting proteins and the dismantling of the railways for intracellular trafficking.

## Data Availability

Not applicable.

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
