# Peer review of "Ubiquitin Proteasome System and Microtubules Are Master Regulators of Central and Peripheral Nervous System Axon Degeneration"

_cells, 2022, doi:10.3390/cells11081358_

Round 1

Reviewer 1 Report

The authors reviewed articles regarding axonal degeneration of the central and peripheral nervous systems by focusing on the roles of microtubule and the ubiquitin-proteasome system. A wide range of issues, including basic aspects and therapeutic applications, are comprehensively covered.

This is an interesting review article shedding light on the significance of microtubule and the ubiquitin-proteasome system in neurodegenerative diseases, such as Parkinson’s disease and Alzheimer’s disease, and peripheral neuropathies, including chemotherapy-induced neuropathy. As these diseases are common in daily clinical practice, this article will attract broad range of readers from basic scientists to physicians. The manuscript is well written, and I do not have any critical comments.

Suggestions to strengthen this manuscript are raised as follows: 

  1. Please reconfirm the use of abbreviations. For example, “PTEN” is spelled out in line 290, whereas it appears in lines 202 and 289 without explanation.
  2. “Microtu-ble-dependent” in line 116 would be “Microtuble-dependent”

Author Response

Thank you for your suggestion. According to the your comments we spelled out PTEN on line 228 and we corrected “Micro-tubule-dependent” in “microtubule-dependent” in the caption of figure 2 (line 129).

"Please see the attachment revised manuscript."

Reviewer 2 Report

This is a review article focused on the relevance of axonal degeneration in in the context of neuronal loss that characterizes neurodegenerative diseases, also discussing similarities and differences with peripheral neuropathies. Although interesting, the manuscript appears poorly focused and it is difficult for the reader to catch the core problem and how axonal degeneration impacts on it.

Major

The introduction section (paragraph1) appears too synthetic, and should be adequately rewritten, adding details and discussion. In this regard, Paragraphs 2-4 could be indicated as sub-chapters of the introduction.

The same comments are true for paragraphs 5 (Parkinson’s disease) and 6 (Peripheral neuropathies).

Paragraph 5 needs a more adequate introduction and a rearrangement of the information there reported. The Authors should better explain why, among several neurodegenerative diseases, they decided to focus on Parkinson’s disease, explaining the relevance of axonal degeneration on the overall scenario of neuronal death. The molecular pathways cited in both paragraphs need a better framing in the general context of molecular features of axon degeneration.

Minor

There are some typos (es. lines 113, 116) and grammar errors (es. Lines 130, 162 and more) that should be amended.

Author Response

Answer to major comments:

We appreciated and accepted your suggestions. Therefore, we divide the introduction in 4 different sub-chapters, and we added some sentences: the first is about programmed cell death and neurodegeneration (lines 35-39); the second aims to explain the choice of the two pathologies (lines 49-57), and two additional references [7.Coleman 2013; 8.Nolano et al., 2017]. In line with this idea, we have rewritten and added some details at the end of chapter 1.4 (lines 170-173). Trying to better focus on our ideas, we added short introductory sentences for both Parkinson‘s Disease (lines 176-181) and peripheral neuropathies (lines 237-241) and two conclusive statements (lines 233-235 and 310-313) in order to better framing our discussion.  In order to widen the horizon, at the end of the discussion (lines 384-3882) we included some evidence about the use of HDAC6 inhibitors in the preclinical model of Charcot-Marie-Tooth neuropathy. In addition, at lines 91 and 248-251 we rewrote a couple of sentences.

Answer to minor comments:

We checked carefully the manuscript and we tried to amend all the typos and grammar errors.

Please see the revised manuscript

Round 2

Reviewer 2 Report

The manuscript is improved and can be accepted for publication even though there are still typos and grammar errors that should be amended.